# A Current-Control Strategy for Grid-Connected Converter Based on Inductance Non-Linear Characteristic Compensation

Xinwen Zhang [1], Shude Yang [2,*] and Yi Zhang [2]

1   School of Electrical and Information Engineering, North Minzu University, Yinchuan 750021, China
2   College of Electrical, Energy and Power Engineering, Yangzhou University, Yangzhou 225000, China
*   Correspondence: yangshude@yzu.edu.cn

**Abstract:** Inductance is a necessary device for a grid-connected converter (GcC) to attenuate the switching-frequency harmonics in injected grid currents. However, in practice, the inductance decreases with an increase in output current. Especially when the amplitude of sinusoidal currents is higher, the inductance will vary over a wide range as the current value changes in a period. This variation may lead to system instability and cause output current fluctuation. To solve this issue, the model of the GcC with a proportional resonant regulator is firstly built, and the system stability with different current values is analyzed using the Nyquist criterion. The results show that the system stability decreases with an increase in current absolute value. Further, a loop gain compensation unit is embedded into the current regulator to maintain the loop gain constant and ensure the stability of the system under a wide variation range of current values. With this scheme, the compensation unit is only determined by the rated value and the non-linear characteristic of the filter inductance. Therefore, the loop gain compensation unit is independent of the original control system, and the traditional controller parameter design method can also be inherited. Finally, the simulated and experimental results from a 50 A static var generator (SVG) with wide filter inductance variation (using Mega-Flux core) have verified the correctness of the analyses and the effectiveness of the proposed method in this paper.

**Keywords:** grid-connected converter; inductance variation; system stability; loop gain; control strategy





## 1. Introduction

Grid-connected converters are broadly employed in power generation from clean energy and power quality regulation fields. Usually, the output filter is necessary to suppress the switching ripple current injected into the grid. In many studies, the inductance value of output filter is regarded as constant [1–4]. However, owing to higher permeability and lower cost, a large number of engineering cases show that when popular materials such as "Fe–Si", "Fe–Si–Al", "Fe–Ni–Mo", and "Fe–Ni" are used for powder core, the inductance value of the output filter will reduce as the absolute value of the inductor current rises [5]. The inductance variation may cause a power quality problem of grid-injected current. Therefore, in order to reduce the low-order harmonics in the current caused by the inductance variation, a proportional–integral (PI) control method based on the combination of a linear and non-linear controller is proposed in [6]. Moreover, a new control strategy based on a proportional resonant (PR) regulator is given in [7]. By using the online adaptive estimation of a non-linear filter inductor, the harmonic compensators' gains of the PR regulator can be reduced, and the acceptable total harmonic distortion (<5%) is achieved.

Except for the power quality problem mentioned above, when the variation in inductance is not considered in controller and circuit parameters design, unstable oscillations in converter output current may also arise [8,9]. Focusing on the unstable oscillations problem, research [10] has found that, with constant control parameters, the converter may become

unstable when the current amplitude increases. Hence, the online parameter tuning idea is used in [11] to handle this problem. However, very accurate current detection between the two pulse-width modulation (PWM) outputs should be ensured in this method. Considering the influence of the non-linear variation in filter inductance on the resonant frequency, the parameter tuning method for LCL filter is given in [12]. Therefore, the conclusions can only be used in the system hardware design stage. Additionally, the influence of inductance variation on switching frequency has been analyzed in [13]. A division–summation (D–Σ) current tuning method is brought in [14] to weaken the influence of broad filter-inductance variety on system stability. In this method, the inductance values under different currents should be measured at the startup for adjusting the loop gain. To cope with the shortages of complex programming, the choice of the region for D–Σ control is unified to general form [15]. Further, the application of D–Σ control is extended to an LCL-filtered GcC to reduce high-frequency ripple current in the grid-injected current [16]. Additionally, in this research, the broad variation in filter inductance has also been considered. Additionally, for the D–Σ control method, accurate current sampling is needed to estimate the non-linear characteristics of filter inductance. To eliminate the influence of inductance variation, i.e., the amplitude of converter output current, on the control bandwidth, a compensation procedure is proposed by linearizing the inductance characteristics [17]. For an LCL-filtered GcC, a renewal direct digital control strategy is put forward to cope with the broad non-linear variation in both grid and inverter side inductances [18]. Furthermore, a coupled analytical method is proposed to reduce the significant discrepancy caused by inductances variation [19]. In the above solutions, the inductance values under different currents should by measured online, and the control system becomes more complex. Furthermore, the noise-sensitive derivative operation on output current is demanded.

On the other hand, in the above methods, the inductance non-linear characteristic compensation algorithm is not independent of the original control system. Because of this, the traditional controller parameter design method may not be suitable for the modified strategy and the design method may need to be updated according to the specific strategy.

In this paper, a control method for a GcC based on inductance non-linear characteristic compensation is proposed. Different from the online non-linear characteristics' estimation method of filter inductance used in the above strategies, this method is based on the curve of inductance vs. the current under offline conditions, which is simple and easy to realize in practice. In this method, only a loop gain compensation unit is embedded into the current controller to conquer the influence of the filter-inductance variety on the loop gain. In particular, the loop gain compensation unit is independent of the original control system, and the traditional controller parameter design method can also be used.

This paper is organized as follows: In Section 2, the mathematical model of the grid-connected converter is derived. Additionally, then the influence of current absolute value on system stability is analyzed in Section 3. In order to deal with the problem of the system stability decreasing with the increase in the current absolute value, an improved control strategy based on the inductance non-linear characteristic compensation is proposed in Section 4. Furthermore, the simulation and experimental results are shown in Section 5 to verify the correctness of the theoretical analysis and the effectiveness of the proposed method. Finally, some conclusions are given in Section 6.

## 2. Mathematical Model of the Grid-Connected Converter

The typical main circuit and control principle of the GcC is shown in Figure 1, where $C$ and $U_{dc}$ represent the dc-bus capacitor and the voltage across it. $u_{inv}$ is the fundamental component of converter output voltage, $L$ is the filter inductance, $i_L$ is the grid-injected current, $u_g$ is the grid voltage, $f$ denotes the grid voltage feedforward path, LPF represents the sampling filter for grid voltage, $i_L^*$ symbolizes the output current reference, $G_i(s)$ stands for the current controller, and $u_{inv}^*$ is the desired converter output voltage.

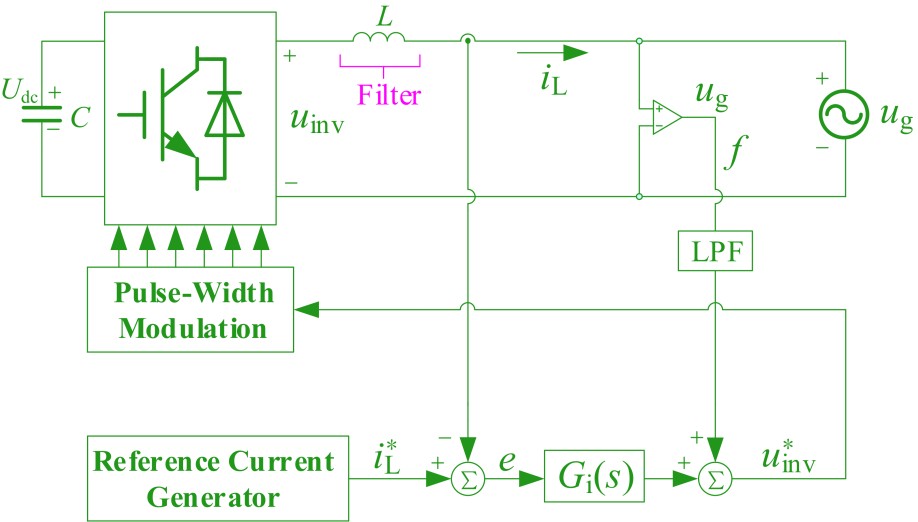

**Figure 1.** The typical main circuit and control principle of GcC.

According to Figure 1, the control schematic of the current loop for the GcC can be acquired, which is exhibited in Figure 2, where $G_F(s)$, $G_d(s)$, and $G_L(s)$ are the transfer function (TF) of the second-order low-pass filter (LPF), control delay and control plant, severally.

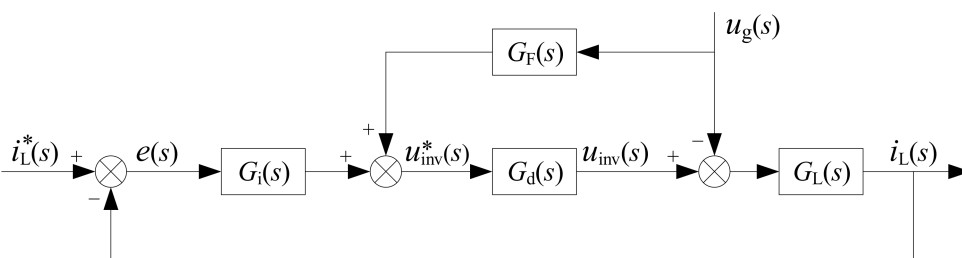

**Figure 2.** Control diagram of the current loop for the GcC.

The current controller has a significant influence on the system performance. In recent years, the PR regulator is widely used in the GcC control due to its strong ability to track the sinusoidal reference without complicated coordinate transformation [20,21], which has a TF of

$$G_i(s) = k_p + \frac{2k_r\omega_c s}{s^2 + 2\omega_c s + \omega_0^2} \tag{1}$$

where $\omega_0$ represents the resonance frequency of PR controller, $k_p$ denotes the proportional part, $k_r$ determines the gain of the controller at resonant frequency and $\omega_c$ provides a bandwidth for the controller to adapt to the grid frequency variation.

The second-order LPF is used here to attenuate high-frequency noises in the grid voltage before it is sampled and used for the phase-locked loop (PLL) and feedforward. The TF of the typical second-order LPF is

$$G_F(s) = \frac{1}{\frac{1}{\omega_b^2}s^2 + \frac{1}{Q\omega_b} + 1} \tag{2}$$

where $\omega_b$ and $Q$ denote the limiting frequency and the quality factor of the second-order LPF, respectively. For the digital controlled GcC system, the delay introduced by computation, sampling, and PWM will exist. When the one-step delay model is used for updating the PWM comparison value, the total delay is about 1.5 control cycles [22], which has a TF of

$$G_d(s) = e^{-1.5T_s s} \tag{3}$$

where $T_s$ denotes the control cycle of the system. From Figure 1, it can be derived that the TF of the control plant is

$$G_L(s) = \frac{1}{Ls} \tag{4}$$

## 3. Analysis of the Influence of Current Absolute Value on System Stability

In practical applications, the inductance value of the output filter is not constant. Usually, the inductance value decreases with an increase in the current absolute value [10,11], and this relation is non-linear. For example, the inductances of the filter with a Mega-Flux core at different current absolute values obtained from the manufacturer are listed in Table 1, and the rated inductance of which is 0.5 mH at 50 A. From Table 1, it is obvious that the inductance of the filter is near 0.5 mH at 50 A. However, when the current through it varies, its inductance value will change correspondingly, i.e., the filter inductance is the function of the current absolute value. Considering this characteristic of the filter, the TF of the control plant shown in (4) can be modified, which is given in (5). If this characteristic of the filter is not considered in the system control, it may affect the system stability and even cause instable oscillation.

$$G_L(s) = \frac{1}{L(|i|)s} \tag{5}$$

**Table 1.** Filter inductances at different current absolute values.

| Symbol | Current Absolute Values and Inductance Values | | | | | | | |
|---|---|---|---|---|---|---|---|---|
| $|I|$ (A) | 0 | 10 | 20 | 30 | 40 | 50 | 60 | 70 |
| $L$ (mH) | 0.71 | 0.69 | 0.67 | 0.62 | 0.56 | 0.48 | 0.41 | 0.34 |

To figure out the influence of current absolute values on the system stability, the open-loop TF of the system can be deduced from Figure 2 when the grid voltage is considered as the disturbance of the control system. Substituting (1), (3) and (5) into the open-loop TF, its expression can be acquired as follows:

$$G_o(s) = G_i(s)G_d(s)G_L(s) = \frac{e^{-1.5T_s s}[k_p s^2 + 2(k_p + k_r)\omega_c s + k_p \omega_0^2]}{L(|i|) \cdot (s^3 + 2\omega_c s^2 + \omega_0^2 s)} \tag{6}$$

The main parameters of the GcC device are shown in Table 2. Based on these parameters and the controller design method given in [23], the $k_p$, $k_r$, $\omega_c$, and $\omega_0$ can be selected as 4, 160, $4\pi$, and $100\pi$, respectively. The unipolar modulation method is used in the system and thus the frequency of the switching ripple will be twice the sampling frequency, which is near 19.2 kHz. In order to restrain the switching components in the grid voltage, the $\omega_b$ and $Q$ are selected as 2 kHz (about one-tenth of the switching ripple frequency) and 0.707, respectively.

Using the above controller parameters, the relation in Table 1, and Formula (6), the Nyquist curves of the open-loop TF of the system under different current absolute values can be acquired, as shown in Figure 3a. From the enlarged views of Figure 3a, it can be known that when the current absolute values are 50 A and 60 A, the crossing points between the Nyquist curves and the real axis are all located to the right of the critical point $(-1,0j)$ which indicates that the GcC is stable. However, when the current absolute value increases to 65 A, the crossing point between the Nyquist curve and the real axis will move to the left of the critical point $(-1,0j)$, which means that the system becomes unstable. Additionally, it shows that when the current absolute value increases to 65 A, the system will be unstable. In this situation, the intersection frequency of the Nyquist curve and the real axis can be obtained as 1500 Hz.

**Table 2.** Main parameters of the grid-tied converter system.

| Parameters | Symbols | Value |
|---|---|---|
| DC-bus capacitor | $C$ | 2820 μF |
| DC-bus voltage | $U_{dc}$ | 400 V |
| Filter inductance | $L$ | 0.5 mH |
| Grid voltage | $u_g$ | 220 V |
| Grid frequency | $f_g$ | 50 Hz |
| Sampling/switching frequency | $f_s$ | 9.6 kHz |

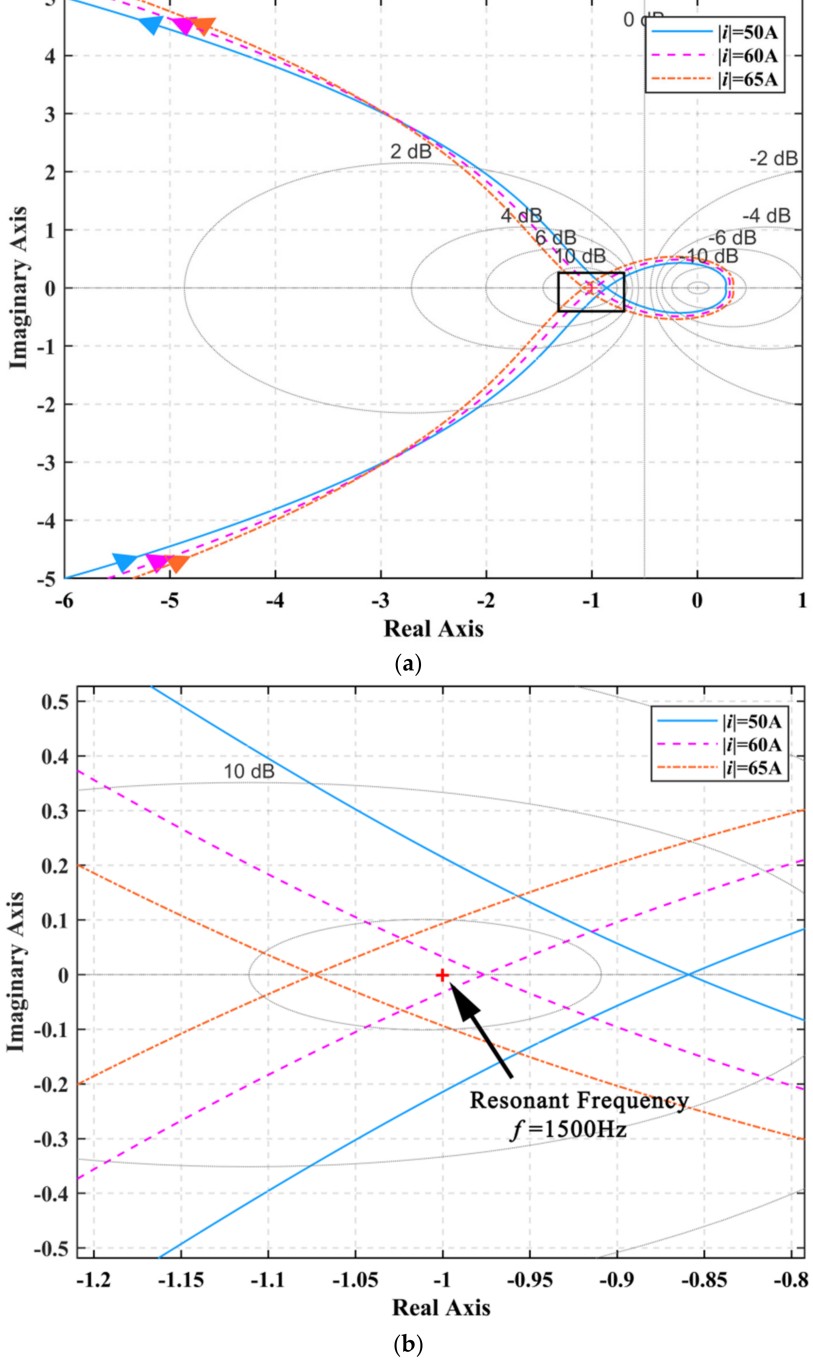

(a)

(b)

**Figure 3.** Stability evaluation of the GcC with different current absolute values: (**a**) overall view of the Nyquist curves when the current absolute value varies; (**b**) enlarged views with different current absolute values.

The above analyses show that the system stability is affected by the current absolute value. Thus, this phenomenon must be treated carefully in the control system design for the GcC. If not, this affect may result in instable oscillations in the converter output current as its absolute value increases.

## 4. Improved Control Strategy Based on the Inductance Non-Linear Characteristic Compensation

From (6), it can be found that the filter inductance value only occurs in the denominator. This means that the loop gain is affected by the filter inductance variation. More specifically, the loop gain of the GcC control system varies and is inversely proportional to the filter inductance. Further, Formula (6) also shows that the loop gain of the GcC control system is in direct proportion to the current controller gain. By combining the above analyses, it can be concluded that the instability problem caused by the filter inductance variation can be solved if the current controller gain is suitably adjusted according to the filter inductance characteristics, i.e., the current absolute values.

In order to meet the adjustment rules of the current controller gain, the characteristics of the filter inductance must be studied. Based on the data shown in Table 1, the Gaussian fitting method is used in this paper to obtain the expression of the filter inductance associated with the current absolute value, which is shown in (7). The fitting result is also shown in Figure 4, and it can be seen that the fitting result matches well with the relationship shown in Table 1.

$$L(|i|) = 0.7115 \times e^{-\left(\frac{|i|-0.8493}{80.74}\right)^2} \tag{7}$$

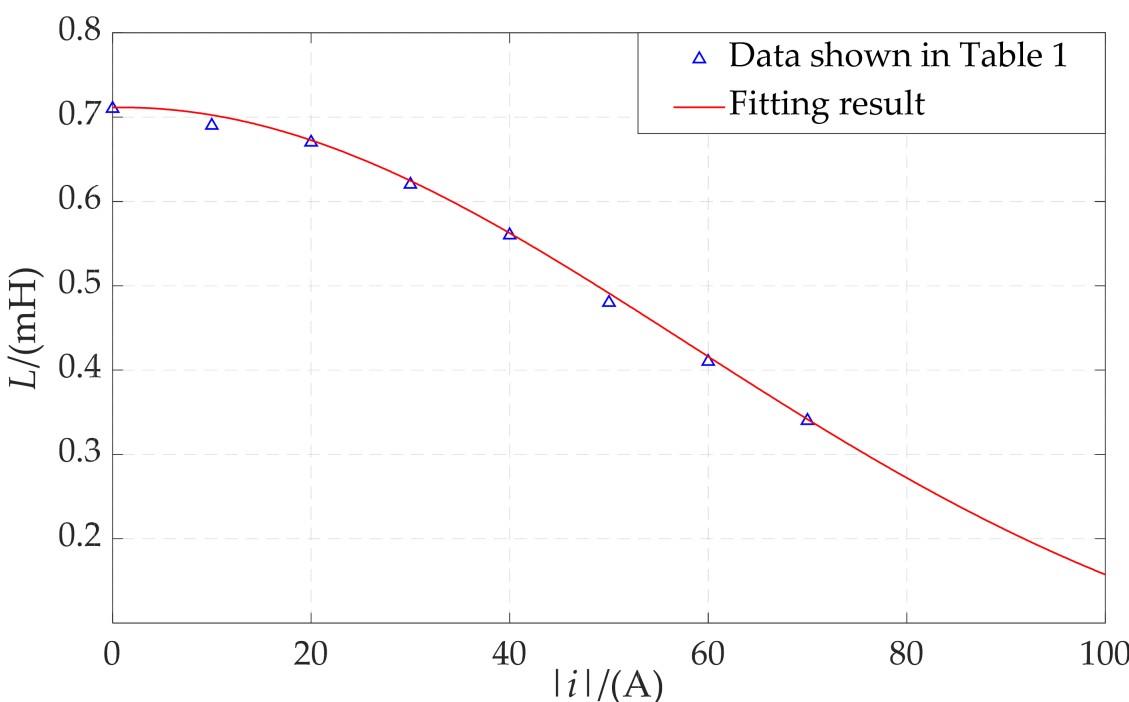

**Figure 4.** Fitting result of the filter inductance characteristics.

It should be noticed that the design of the controller parameters is based on the rated inductance value of the filter. Naturally, in this situation, the loop gain is suitable for the expected system performance, which is denoted by $G_{o\_designed}(s)$. Thus, if the loop can always be equal to the $G_{o\_designed}(s)$, the performance and the stability of the GcC system will not be affected by the variation in current values. In order to realize this assumption, a control strategy of the GcC based on the inductance non-linear characteristic compensation is proposed in this paper, and the control chart of which is shown in Figure 5. In the proposed control strategy, a loop gain compensation unit which is expressed by $K(|i|)$ is

embedded into the current regulator, and its gain is calculated using (8). From Figure 5 and Formula (8), it can be found that $K(|i|)$ is only determined by the rated value and the non-linear characteristic of the filter inductance. This means that the loop gain compensation unit is independent of the original control system, and the traditional controller parameter design method can also be used.

$$K(|i|) = \frac{L(|i|)}{L_{\text{rated}}} \tag{8}$$

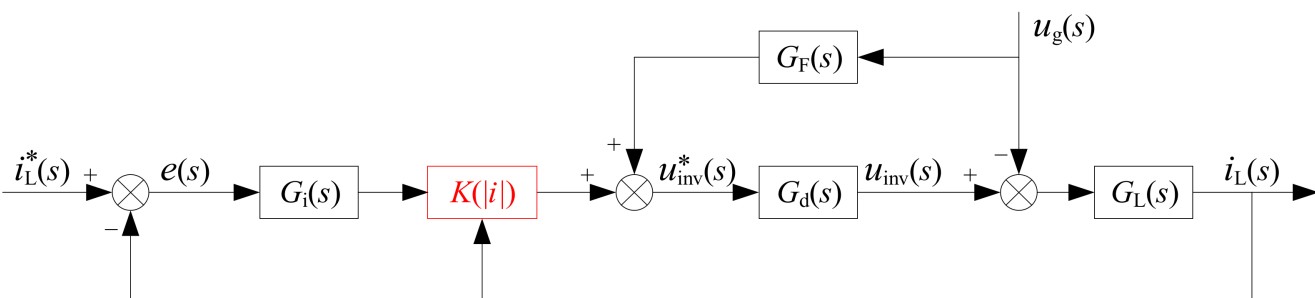

**Figure 5.** Control chart of the current loop based on the inductance non-linear characteristic compensation.

From Figure 5, it can be derived that when the loop gain compensation unit is introduced, the open-loop TF is modified as follows

$$G_{\text{o}}(s) = G_{\text{i}}(s)G_{\text{d}}(s)G_{\text{L}}(s)K(|i|) = \frac{K(|i|)e^{-1.5T_{\text{s}}s}[k_{\text{p}}s^2 + 2(k_{\text{p}} + k_{\text{r}})\omega_{\text{c}}s + k_{\text{p}}\omega_0^2]}{L(|i|) \cdot (s^3 + 2\omega_{\text{c}}s^2 + \omega_0^2 s)} \tag{9}$$

Considering the expression in (8), the open-loop TF will be simplified as

$$G_{\text{o}}(s) = G_{\text{i}}(s)G_{\text{d}}(s)G_{\text{L}}(s)K(|i|) = \frac{e^{-1.5T_{\text{s}}s}[k_{\text{p}}s^2 + 2(k_{\text{p}} + k_{\text{r}})\omega_{\text{c}}s + k_{\text{p}}\omega_0^2]}{L_{\text{rated}} \cdot (s^3 + 2\omega_{\text{c}}s^2 + \omega_0^2 s)} \tag{10}$$

From (10), it is obvious that when the loop gain compensation unit is introduced, the loop gain of the control system will only be associated with the rated value of filter inductance, which means that the performance and the stability of the GcC system will not be affected by the filter inductance variation, i.e., the current variation. With the control strategy based on the inductance non-linear compensation, the Nyquist curves of the open-loop TF under different current absolute values can be obtained, as shown in Figure 6a. The enlarged views of Figure 6a show that when the current absolute values are 50 A, 60 A, and 65 A, the intersection points between the Nyquist curves and the real axis will remain unchanged. Therefore, the influence caused by the variation in the filter inductance on the system stability is eliminated.

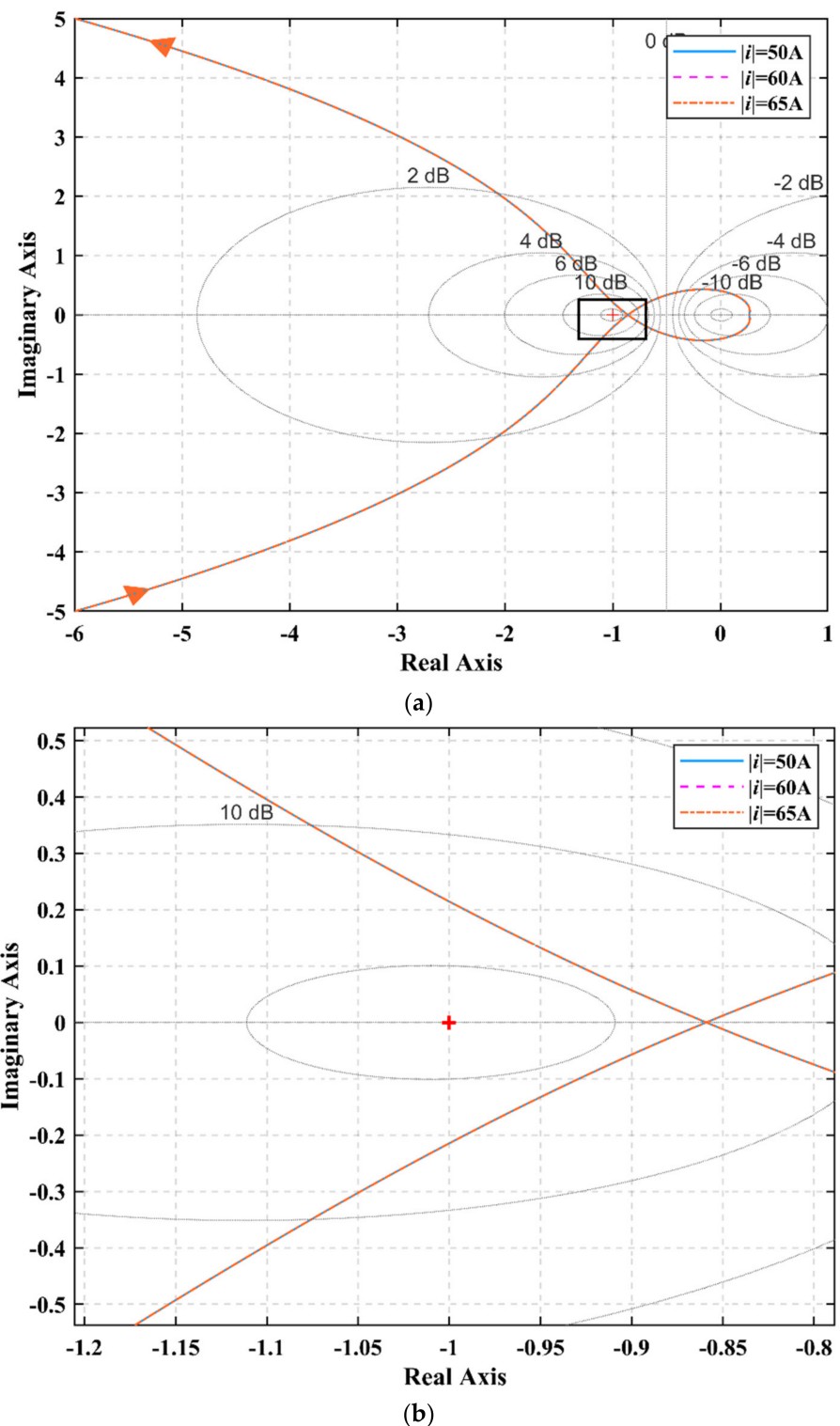

**Figure 6.** Stability evaluation of the GcC with different current absolute values for the proposed control strategy: (**a**) overall view of the Nyquist curves when the current absolute value varies; (**b**) enlarged views with different current absolute values.

## 5. Simulation and Experimental Results

In order to demonstrate the validity of the above theoretical analyses and the effectiveness of the proposed method, the simulations and experiments are performed using PLECS standalone 4.1.2 and the prototype. The characteristic of the filter inductance is shown in Table 2.

### 5.1. Simulation Results

In simulation, the "variable inductor" component in PLECS is used to construct the filter inductance with the characteristics shown in Table 2. The simulation results with different magnitudes of output current reference before the loop gain compensation unit is introduced are illustrated in Figure 7. Figure 7a shows that when the magnitude of output current reference is 60 A, the output current has good sinusoidal. Therefore, the GcC is stable in this situation. However, Figure 7b shows that when the magnitude of output current reference is up to 70 A, the oscillations occur near the peak value of the current waveforms and the system becomes unstable. The reason for this phenomenon is that the stability of the system decreases with an increase in the current absolute value. Further, Figure 7c shows that when the system is unstable, the oscillation frequency is 1500 Hz. These simulation results match well with the analyses in Section 3.

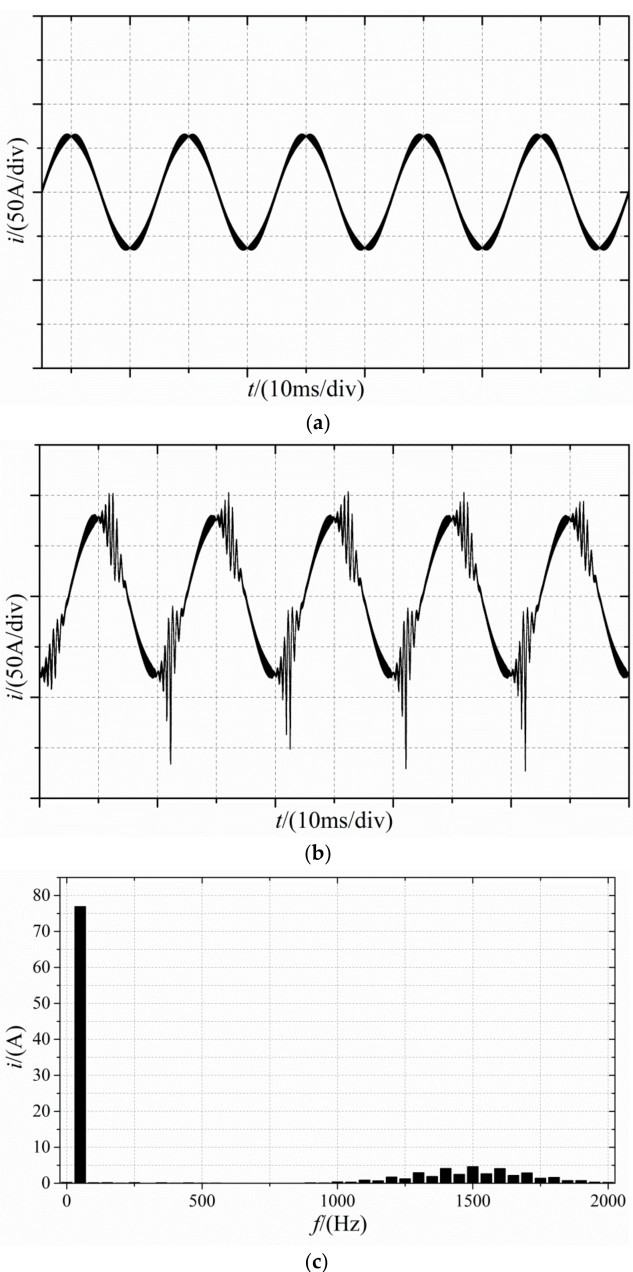

**Figure 7.** Simulation results with different magnitudes of output current reference before the loop gain compensation unit is introduced: (**a**) magnitude of output current reference is 60 A; (**b**) magnitude of output current reference is 70 A; (**c**) spectra of the output current when its reference magnitude is 70 A.

For comparison, Figure 8 gives the simulation results with different magnitudes of output current reference when the loop gain compensation unit is introduced. Obviously, the GcC system is always stable when the magnitudes of output current reference are 60 A and 70 A, respectively. The reason for this phenomenon is that the loop gain of the system can remain constant owing to the loop gain compensation unit, even though the filter inductance, i.e., the current absolute value, varies widely. These simulation results are corresponded well with the analyses in Section 4, which verifies the effectiveness of the proposed control strategy of the GcC based on inductance non-linear characteristic compensation.

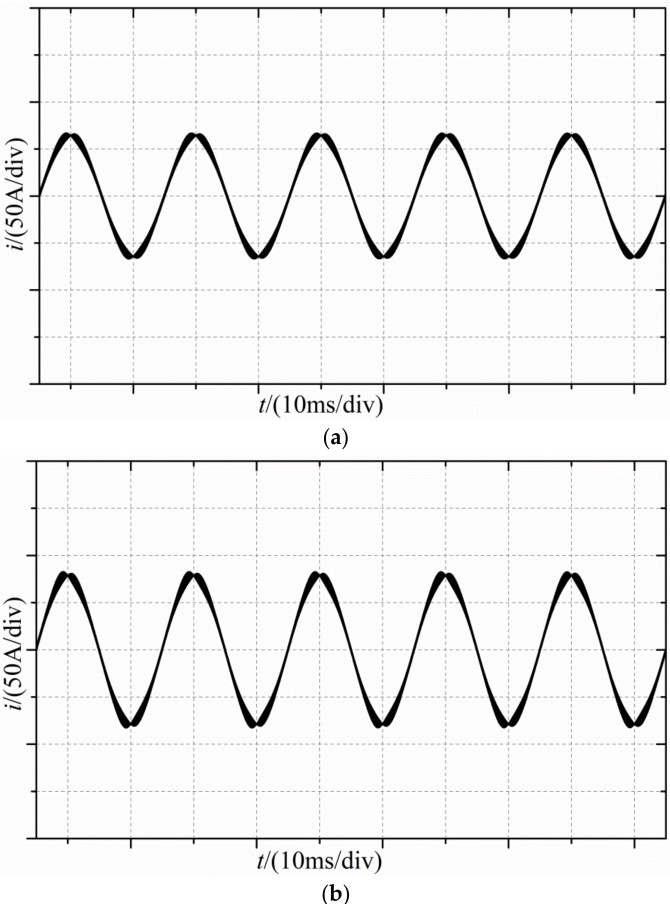

**Figure 8.** Simulation results with different magnitudes of output current reference when the loop gain compensation unit is introduced: (**a**) magnitude of output current reference is 60 A; (**b**) magnitude of output current reference is 70 A.

### 5.2. Experimental Results

To attest the accuracy of the analyses and the effectiveness of the proposed control strategy in this paper, an experimental prototype is designed, which is shown in Figure 9. The control board in the experimental system mainly consists of a current and voltage measurement circuit, a fault detection and protection circuit, and TMS320F28335 DSP. The touch panel is used for man–machine interaction to give the start or stop command and adjust the magnitude of the output current. The Mega-Flux core is used for the filter inductance in the experiment, and the characteristics of the filter inductance are displayed in Table 2.

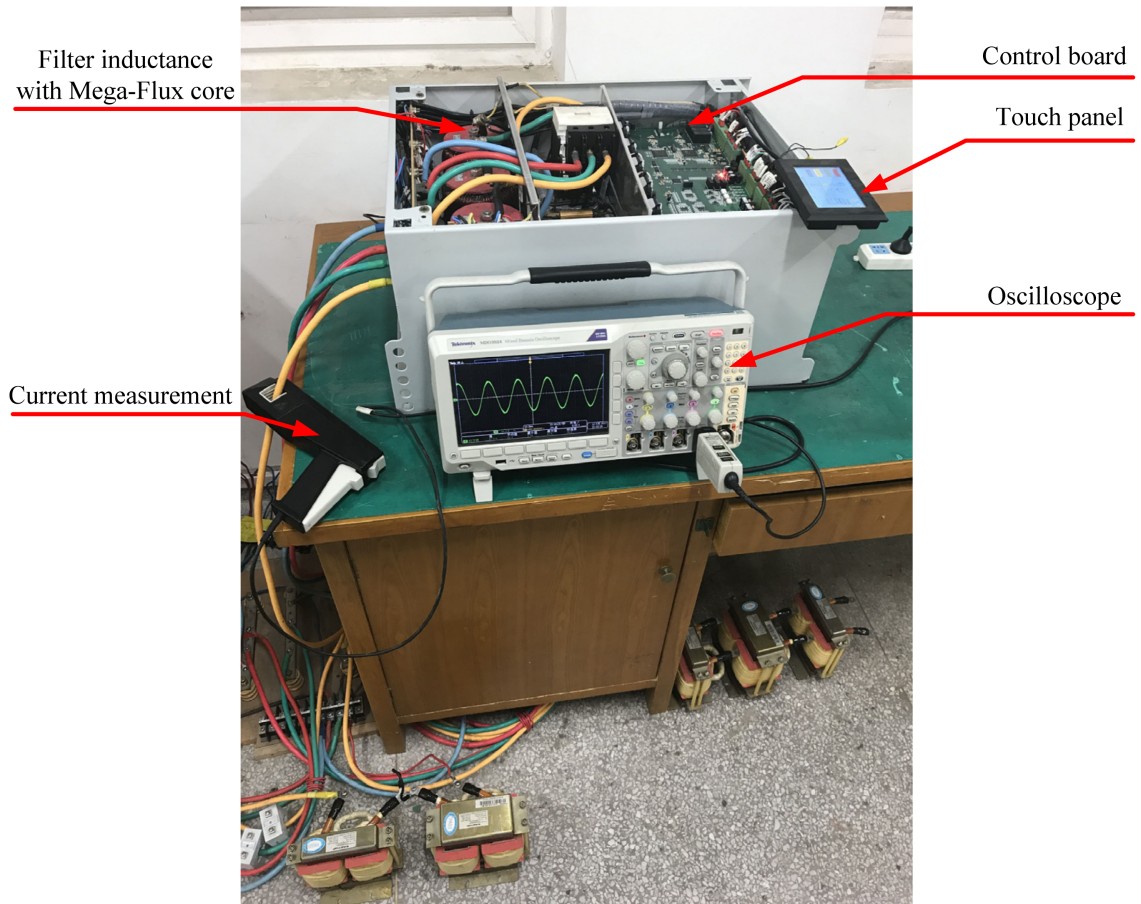

**Figure 9.** Experimental prototype of the GcC.

The experimental results with different magnitudes of output current reference before the loop gain compensation unit is introduced are shown in Figure 10. Figure 10a describes that when the magnitude of output current reference is 60 A, the GcC is stable. However, Figure 10b shows that when the magnitude of output current reference is up to 70 A, the oscillations occur near the peak value of the current waveforms, and the system becomes unstable. Further, Figure 10c shows that when the system is unstable, the oscillation frequency is 1500 Hz. These experimental results are consistent with the analyses in Section 3 and the above simulation results.

In addition, Figure 11 gives the experimental results with different magnitudes of output current reference when the proposed control strategy is used. Obviously, the GcC always keeps stable when the magnitudes of output current reference are 60 A and 70 A, respectively. These simulation results match well with the analyses in Section 4 and the above simulation results.

From the above results obtained from the simulation and experiment, it can be concluded that the proposed control strategy of the GcC based on inductance non-linear characteristic compensation is effective to guarantee the stability of the system under wide-range variation in current values.

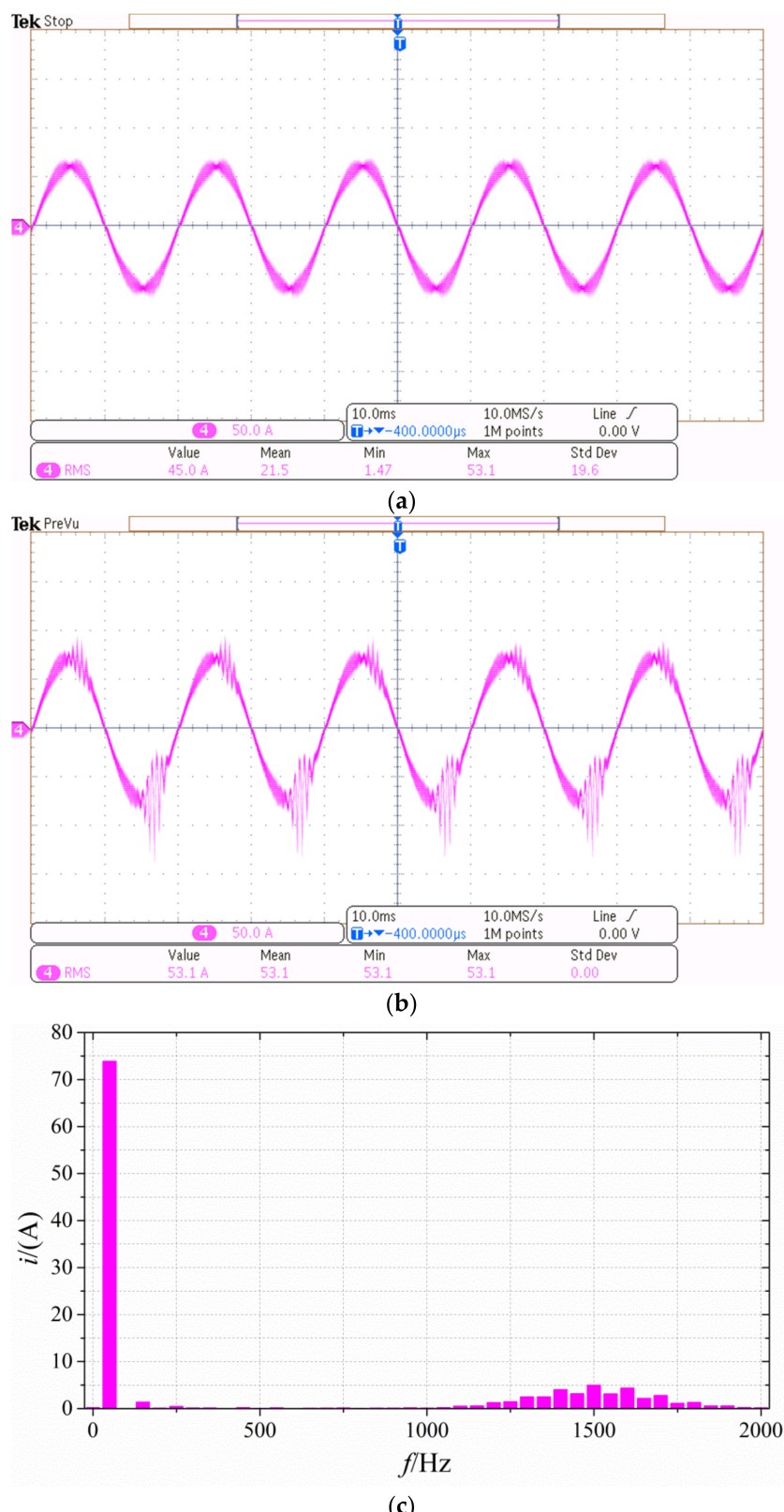

**Figure 10.** Experimental results with different magnitudes of output current reference before the loop gain compensation unit is introduced: (**a**) magnitude of output current reference is 60 A; (**b**) magnitude of output current reference is 70 A; (**c**) spectra of the output current when its reference magnitude is 70 A.

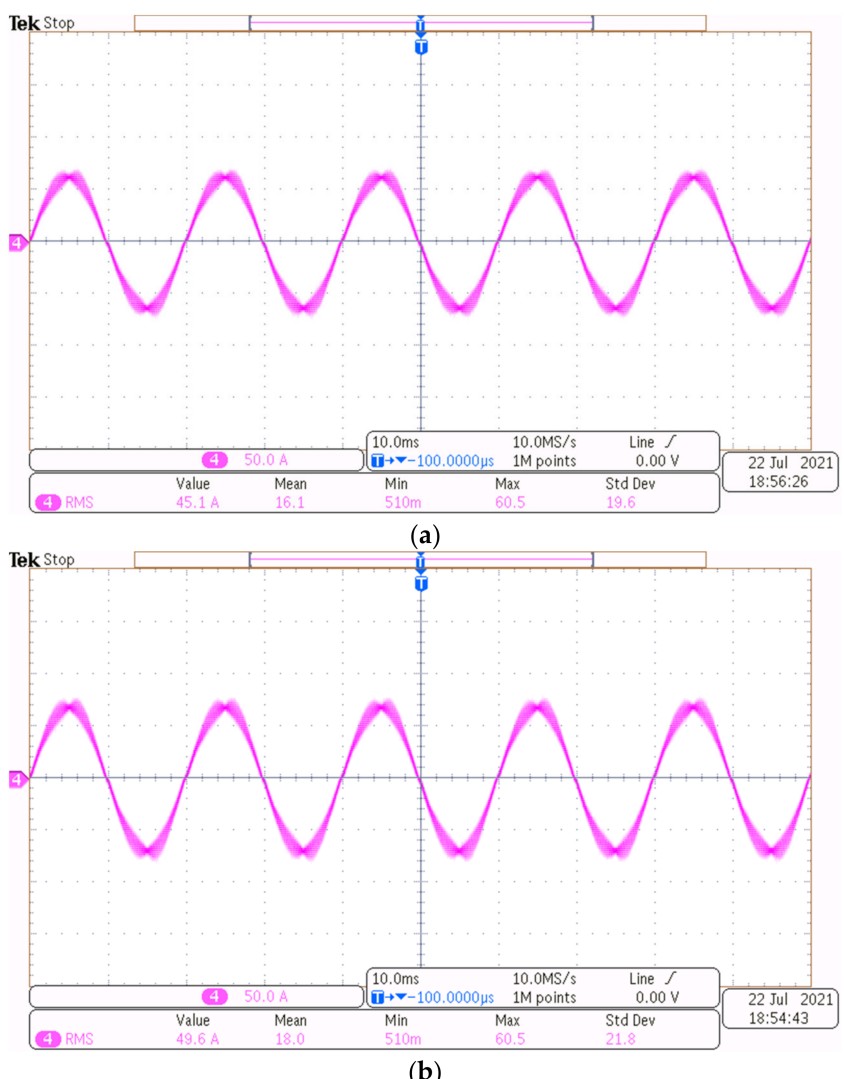

**Figure 11.** Experimental results with different magnitudes of output current reference when the loop gain compensation unit is introduced: (**a**) magnitude of output current reference is 60 A; (**b**) magnitude of output current reference is 70 A.

## 6. Conclusions

The aim of this paper was to solve the stability problem of the GcC system caused by the wide-range variation in filter inductance values, i.e., current absolute values. Through analyzing the system stability under different current absolute values, it was revealed that the system stability decreases with the growth of current absolute value. For a GcC system with a 0.5-mH-rated inductance at 50 A, the system will become unstable and the oscillations will occur near the peak value of the current waveforms when the magnitude of output current reference is 70 A. Additionally, the oscillation frequency was near 1500 Hz. The reason for this problem is that the loop gain is affected by the filter inductance variation. If this problem is not considered in the control system design, it may lead to instability and the oscillations will occur near the peak value of the current waveforms. To eliminate the adverse effect of the filter inductance variation on the system stability, an improved control strategy based on inductance non-linear characteristic compensation is proposed, in which a loop gain compensation unit is embedded into the current regulator to maintain the loop gain constant. In this regard, the loop gain will only be associated with the rated inductance value of the output filter. Therefore, the stability of the system under wide-range variation in current values can be ensured. In addition, the compensation unit is only determined by the rated value and the non-linear characteristic of the filter inductance. Therefore, the loop

gain compensation unit is independent of the original control system, and the traditional controller parameter design method can also be inherited.

The main limitation of the proposed method is that the non-linear characteristic of the filter inductance needs to be obtained from the manufacturer. When the filter inductance is replaced, the non-linear characteristic compensation unit must be updated manually. Therefore, in our future work the learning algorithm is considered to operate at the beginning of the GcC startup to update the non-linear characteristic of the filter inductance.

**Author Contributions:** Conceptualization, S.Y. and X.Z.; methodology, X.Z.; software, Y.Z.; validation, Y.Z.; formal analysis, X.Z.; investigation, S.Y.; resources, Y.Z.; data curation, S.Y.; writing—original draft preparation, S.Y.; writing—review and editing, X.Z.; visualization, S.Y.; supervision, S.Y.; project administration, X.Z.; funding acquisition, X.Z. All authors have read and agreed to the published version of the manuscript.

**Funding:** This research was funded by the Natural Science Research of Jiangsu Higher Education Institutions of China (grant number: 19KJB470038), National Nature Science Foundation of China (grant number: 51867001) and Ningxia Natural Science Foundation (grant number: 2020AAC03210).

**Conflicts of Interest:** The authors declare no conflict of interest.

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
