# Peer review of "A Current-Control Strategy for Grid-Connected Converter Based on Inductance Non-Linear Characteristic Compensation"

_electronics, doi:10.3390/electronics11244170_

Round 1
Reviewer 1 Report
1- I recommend the authors improve the paper title such as removing "Research on.. " from title
2- It could be interesting to summarize the commented literature works in a table to have a clear comparison between all. This could also help precisely formulating the contribution of the paper with respect to previous works
3- The research paper should be written in the perspective of the third person. Words such as ‘I’ ‘we’ ‘our’ etc. needs to be avoided. Such as: " we proposed a novel control strategy....",
4- In line 281, the authors claim to suggest a novel mechanism operates on the MG through the sentence " a novel control strategy….", But from my experience, I did not find a new mechanism in this paper. What are the contributions of the paper?
5- The results are not stable and contain high noise as " Figure 10. Experimental results with different magnitudes of output current before the loop gain compensation unit is introduced: (b) magnitude of output ......", therefore all results need improving
6- I recommend the authors support the conclusions section by results
· Domenico SgròWilkley Bezerra CorreiaWilkley Bezerra CorreiaRuth LeaoRuth Leao, Silas Alysson Souza Tibúrcio, "Nonlinear current control strategy for grid-connected voltage source converters ", International Journal of Electrical Power & Energy Systems 142(3):108349, DOI: 10.1016/j.ijepes.2022.108349
· Bilal Naji Alhasnawi, Basil H Jasim, Walid Issa, M Dolores Esteban, "A novel cooperative controller for inverters of smart hybrid AC/DC microgrids ", Appl. Sci. 2020, 10(17), 6120; https://doi.org/10.3390/app10176120
· Tooraj Abbasian Najafabadi Farzad Rajaei Salmasi Hossein Safamehr Hossein Safamehr, "Enhanced control of grid-connected inverters with non-linear inductor in LCL filter ", IET Power Electronics 9(10), DOI: 10.1049/iet-pel.2015.0312
Author Response
Dear Editors and Reviewers:
Thank you for your letter and for the reviewer’s positive and constructive comments concerning our manuscript. These comments are all valuable and very helpful for revising and improving our paper, as well as the important guiding significance to our future researches. We have discussed on these comments carefully and have made corrections and revisions that are marked in red in the revised manuscript. We really appreciate the comments from the reviewers and hope our response may give satisfactory feedback. Our responses to several comments are listed below:
Comment 1:I recommend the authors improve the paper title such as removing "Research on.. " from title
Replay: Thank you very much for your comment. The paper title has been improved according to your comment.
Comment 2:It could be interesting to summarize the commented literature works in a table to have a clear comparison between all. This could also help precisely formulating the contribution of the paper with respect to previous works
Replay: Thank you very much for your comment. Different from the online non-linear characteristics estimation method of filter inductance used in the commented strategies, the proposed method is based on the curve of inductance vs. current under offline conditions, which is simple and easy to realize in practice. In the proposed method, only a loop gain compensation unit is embedded into the current controller to conquer the influence of the filter-inductance variety on the loop gain. Especially, the loop gain compensation unit is independent of the original control system, and the traditional controller parameter design method can also be used. These comparisons have been added and marked in red in the Introduction Section of the revised manuscript.
Comment 3:The research paper should be written in the perspective of the third person. Words such as ‘I’ ‘we’ ‘our’ etc. needs to be avoided. Such as: " we proposed a novel control strategy....",
Replay: Thank you very much for your comment. We have checked the paper and modified the description method.
Comment 4:In line 281, the authors claim to suggest a novel mechanism operates on the MG through the sentence " a novel control strategy….", But from my experience, I did not find a new mechanism in this paper. What are the contributions of the paper?
Replay: Thank you very much. I’m sorry that we failed to highlight the contributions in our work. The aim of the paper is to solve the stability problem of the grid-connected converter system caused by the wide range variation of filter inductance value, i.e., the current absolute value. Compared with the other strategies, the main contributions of the paper can be summarized:
(1) Different from the online non-linear characteristics estimation method of filter inductance used in the commented strategies, the proposed method is based on the curve of inductance vs. current under offline conditions, which is simple and easy to realize in practice.
(2) In the proposed method, the loop gain compensation unit is only determined by the rated value and the non-linear characteristic of the filter inductance. Therefore, the loop gain compensation unit is independent of the original control system, and the traditional controller parameter design method can also be used.
These comparisons have been added and marked in red in the revised manuscript and "novel" has been replaced by "improved" for more precise description.
Comment 5:The results are not stable and contain high noise as " Figure 10. Experimental results with different magnitudes of output current before the loop gain compensation unit is introduced: (b) magnitude of output ......", therefore all results need improving
Replay: Thank you very much for your comment. According to the theoretical analysis in Section 3, the unstable results in Figure 10 are caused by the non-linear characteristics of filter inductance. And the reason for the appearance of "high noise" is the unstable fluctuation. The aim of this paper is to solve this problem. As you said, when the output current is unstable, "magnitudes of output current" will be not accurate enough to describe the output current. According to your comments, "magnitudes of output current" has been replaced by "magnitudes of output current reference" in the revised manuscript.
Comment 6:I recommend the authors support the conclusions section by results
Replay: Thank you very much for your comment. The conclusions given in this paper are supported by theoretical analysis, simulation and experimental results. The details are as follows:
Through analyzing the system stability under different current absolute values, it can be revealed that the system stability decreases with the growth of current absolute value. The reason for this problem is that the loop gain is affected by the filter inductance variation. If this problem is not considered in the control system design, it may lead to instability and the oscillations will occur near the peak value of the current waveforms (supported by theoretical analysis in Figure 3, simulation results in Figure 7 and experimental results in Figure 10). To eliminate the adverse effect of the filter inductance variation on the system stability, an improved control strategy based on inductance non-linear characteristic compensation is proposed, in which, a loop gain compensation unit is embedded into the current regulator to maintain the loop gain constant. In this way, the loop gain will be only associated with the rated inductance value of the output filter. Therefore, the stability of the system under wide range variation of current values can be ensured (supported by theoretical analysis in Figure 6, simulation results in Figure 8 and experimental results in Figure 11).
Yours sincerely,
The authors.

Reviewer 2 Report
This paper is about Research on Control Strategy for Grid-Connected Converter Based on Inductance Non-Linear Characteristic Compensation. The paper is nicely written, and I have the following comments:
1 – The motivation in Section 1 is weak. Please motivate your work.
2 – At the end of Section 1, write a paragraph to summarize the paper. Example, the paper is structured as follows: Sections 2 presents ….. etc.
3 – As we know, non-linear systems are more complicated than linear ones. What is the complexity of the proposed model in comparison with similar models? I mean it is a trade off between accuracy and complexity. We need models to give higher accuracy but at the same time we do not want the complexity to be very high
4 – Related studies section is missing.
5 – How are the results of the proposed work compared to related work?
6 – What are the main limitations of the proposed model? You can a paragraph in Section 6 to discuss the limitations.
7 – Also, in Section 6, add a paragraph to discuss future work
Author Response
Dear Editors and Reviewers:
Thank you for your letter and for the reviewer’s positive and constructive comments concerning our manuscript. These comments are all valuable and very helpful for revising and improving our paper, as well as the important guiding significance to our future researches. We have discussed on these comments carefully and have made corrections and revisions that are marked in red in the revised manuscript. We really appreciate the comments from the reviewers and hope our response may give satisfactory feedback. Our responses to several comments are listed below:
Comment 1:The motivation in Section 1 is weak. Please motivate your work.
Replay: Thank you very much for your comment. According to your comment, the motivation of our work has been improved, which are added and marked in red in the Introduction section of the revised manuscript.
Comment 2:At the end of Section 1, write a paragraph to summarize the paper. Example, the paper is structured as follows: Sections 2 presents ….. etc.
Replay: Thank you very much for your comment. According to your comment, a paragraph to summarize the paper has been given in the end of Section 1.
Comment 3:As we know, non-linear systems are more complicated than linear ones. What is the complexity of the proposed model in comparison with similar models? I mean it is a trade off between accuracy and complexity. We need models to give higher accuracy but at the same time we do not want the complexity to be very high
Replay: Thank you very much for your comment. This is a very important question. The model in this paper is actually the description of the nonlinear system under different steady-state operating points. Owing to this process, the nonlinear model of the system can be transformed to a linear one. Thereby, the complexity of the system model can be reduced and the analysis theory of linear system, such as Nyquist criterion can be used to analyze the stability of nonlinear system. Of course, as you said, the linearized model cannot fully reflect the behavior of the nonlinear systems. However, the focus of this paper is not to accurately measure the relative stability of the system, but to pay attention to the variation trend of the system stability. The simulation and experimental results show that in this research, the above linearized model is acceptable.
Comment 4:Related studies section is missing.
Replay: Thank you very much for your comment. The related studies have been added to Section 1.
Comment 5:How are the results of the proposed work compared to related work?
Replay: Thank you very much for your comment. The main contributions of the paper can be summarized:
(1) Different from the online non-linear characteristics estimation method of filter inductance used in the commented strategies, the proposed method is based on the curve of inductance vs. current under offline conditions, which is simple and easy to realize in practice.
(2) In the proposed method, the loop gain compensation unit is only determined by the rated value and the non-linear characteristic of the filter inductance. Therefore, the loop gain compensation unit is independent of the original control system, and the traditional controller parameter design method can also be used.
These comparisons have been added and marked in red in the revised manuscript.
Comment 6:What are the main limitations of the proposed model? You can a paragraph in Section 6 to discuss the limitations.
Replay: Thank you very much for your comment. The main limitation of the proposed method is that the non-linear characteristic of the filter inductance needs to be obtained from the manufacturer. When the filter inductance is replaced, the non-linear characteristic compensation unit must be updated manually. The above descriptions have been added and marked in red in Section 6.
Comment 7:Also, in Section 6, add a paragraph to discuss future work
Replay: Thank you very much for your comment. A paragraph to discuss future work have been added and marked in red in Section 6.
Yours sincerely,
The authors.

Reviewer 3 Report
The topic of the paper is interesting and fits the scopes of the Journal. The manuscript requires some extra efforts to improve its quality and presentation. After a careful revision, a set of comments are given below.
The term “control strategy” could be added as keyword, if the authors agree.
The last paragraph of the Introduction should be rewritten in order to include the section numbers where the contents are found. In other words, in scientific paper it is a common to include a few lines describing the structure of the rest of the manuscript. This aspect would help to enhance the readability.
In the legend of figure 4, the initial “The” of “The data shown in Table 1” should be removed from this humble reviewer opinion.
In section 5 it is indicated that PLECS is used to perform simulations; however, the specific version that has been used is not mentioned. It suggested including this detail for a proper description.
Figure 9 shows the experimental setup. In this sense, the components Control board and Touch panel are only seen in the photograph, they do not appear within the previous text. Therefore, it is suggested to mention them, at least in a brief manner.
The main limitations of the work should be commented in a brief manner for a better presentation. This could be done in the last section.
Future research guidelines should be briefly mentioned in the conclusions section.
It must be remarked that from this reviewer viewpoint, reporting experimental results is a positive feature of the paper.
Author Response
Dear Editors and Reviewers:
Thank you for your letter and for the reviewer’s positive and constructive comments concerning our manuscript. These comments are all valuable and very helpful for revising and improving our paper, as well as the important guiding significance to our future researches. We have discussed on these comments carefully and have made corrections and revisions that are marked in red in the revised manuscript. We really appreciate the comments from the reviewers and hope our response may give satisfactory feedback. Our responses to several comments are listed below:
Comment 1:The term “control strategy” could be added as keyword, if the authors agree.
Replay: Thank you very much for your comment. According to your comment, the term “control strategy” has been added as keyword, which is marked in red in the revised manuscript.
Comment 2:The last paragraph of the Introduction should be rewritten in order to include the section numbers where the contents are found. In other words, in scientific paper it is a common to include a few lines describing the structure of the rest of the manuscript. This aspect would help to enhance the readability.
Replay: Thank you very much for your comment. According to your comment, a paragraph to summarize the paper has been given in the end of Section 1.
Comment 3:In the legend of figure 4, the initial “The” of “The data shown in Table 1” should be removed from this humble reviewer opinion.
Replay: Thank you very much for your comment. According to your comment, the initial “The” of “The data shown in Table 1” in the legend of figure 4 has been removed.
Comment 4:In section 5 it is indicated that PLECS is used to perform simulations; however, the specific version that has been used is not mentioned. It suggested including this detail for a proper description.
Replay: Thank you very much for your comment. The specific version of PLECS software has been given and marked in red in the revised manuscript.
Comment 5:Figure 9 shows the experimental setup. In this sense, the components Control board and Touch panel are only seen in the photograph, they do not appear within the previous text. Therefore, it is suggested to mention them, at least in a brief manner.
Replay: Thank you very much for your comment. The control board in experimental system mainly consists of current and voltage measurement circuit, fault detection and protection circuit and TMS320F28335 DSP. The touch panel is used for man-machine interaction to give the start or stop command and adjust the magnitude of output current. The above descriptions have been added and marked in red in Section 6 in the revised manuscript.
Comment 6:The main limitations of the work should be commented in a brief manner for a better presentation. This could be done in the last section.
Replay: Thank you very much for your comment. The main limitation of the proposed method is that the non-linear characteristic of the filter inductance needs to be obtained from the manufacturer. When the filter inductance is replaced, the non-linear characteristic compensation unit must be updated manually. The above descriptions have been added and marked in red in the last Section.
Comment 7:Future research guidelines should be briefly mentioned in the conclusions section.
Replay: Thank you very much for your comment. A paragraph to discuss the future work have been added and marked in red in the conclusions Section.
Yours sincerely,
The authors.

Round 2
Reviewer 1 Report
1- The authors must be supports the conclutions by results
2- the authors must be supports the abstract by results
3- The authors may enrich their references with the latest and related work further, such as below, and more:
• Domenico SgròWilkley Bezerra CorreiaWilkley Bezerra CorreiaRuth LeaoRuth Leao, Silas Alysson Souza Tibúrcio, "Nonlinear current control strategy for grid-connected voltage source converters ", International Journal of Electrical Power & Energy Systems 142(3):108349, DOI: 10.1016/j.ijepes.2022.108349
• Bilal Naji Alhasnawi, Basil H Jasim, Walid Issa, M Dolores Esteban, "A novel cooperative controller for inverters of smart hybrid AC/DC microgrids ", Appl. Sci. 2020, 10(17), 6120; https://doi.org/10.3390/app10176120
• Tooraj Abbasian Najafabadi Farzad Rajaei Salmasi Hossein Safamehr Hossein Safamehr, "Enhanced control of grid-connected inverters with non-linear inductor in LCL filter ", IET Power Electronics 9(10), DOI: 10.1049/iet-pel.2015.0312
Author Response
Dear Editors and Reviewers:
We would like to extend our heartfelt thanks to you for your great attention paid to our paper. Again very many thanks to you for your valuable comments and suggestions. We have made our great efforts to revise what you have pointed out in accordance with your suggestion. And now, we would like to have our paper at your disposal, hoping that the paper will be published in your journal. Our responses to several comments are listed below:
Comment 1:The authors must be supports the conclutions by results
Replay: Thank you very much for your comment. The conclusions have been improved by adding the simulation and experimental results. The revisions made to the manuscript are marked in red in the revised manuscript.
Comment 2:the authors must be supports the abstract by results
Replay: Thank you very much for your comment. The abstracts have been modified by adding the theoretical analysis results. The revisions made to the manuscript are marked in red in the revised manuscript.
Comment 3:The authors may enrich their references with the latest and related work further, such as below, and more:
- Domenico SgròWilkley Bezerra CorreiaWilkley Bezerra CorreiaRuth LeaoRuth Leao, Silas Alysson Souza Tibúrcio, "Nonlinear current control strategy for grid-connected voltage source converters ", International Journal of Electrical Power & Energy Systems 142(3):108349, DOI: 10.1016/j.ijepes.2022.108349
- Bilal Naji Alhasnawi, Basil H Jasim, Walid Issa, M Dolores Esteban, "A novel cooperative controller for inverters of smart hybrid AC/DC microgrids ", Appl. Sci. 2020, 10(17), 6120; https://doi.org/10.3390/app10176120
- Tooraj Abbasian Najafabadi Farzad Rajaei Salmasi Hossein Safamehr Hossein Safamehr, "Enhanced control of grid-connected inverters with non-linear inductor in LCL filter ", IET Power Electronics 9(10), DOI: 10.1049/iet-pel.2015.0312
Replay: Thank you very much for your comment. The above three references and their contributions have been added to the revised manuscript, which are marked in red in the revised manuscript.
Yours sincerely,
The authors.

Reviewer 2 Report
The authors addressed my comments
Author Response
Dear Editors and Reviewers:
We would like to extend our heartfelt thanks to you for your great attention paid to our paper. Again very many thanks to you for your valuable comments and suggestions. And now, we hope that the paper will be published in your journal.
Yours sincerely,
The authors.
